# Don't Start From Scratch: Leveraging Prior Data to Automate Robotic Reinforcement Learning

Homer Walke[1], Jonathan Yang[1], Albert Yu[2], Aviral Kumar[1], Jędrzej Orbik[1], Avi Singh[3], and Sergey Levine[1,3]

[1]UC Berkeley  [2]UT Austin  [3]Google

**Abstract:** Reinforcement learning (RL) algorithms hold the promise of enabling autonomous skill acquisition for robotic systems. However, in practice, real-world robotic RL typically requires time consuming data collection and frequent human intervention to reset the environment. Moreover, robotic policies learned with RL often fail when deployed beyond the carefully controlled setting in which they were learned. In this work, we study how these challenges can all be tackled by effective utilization of diverse offline datasets collected from previously seen tasks. When faced with a new task, our system adapts previously learned skills to quickly learn to both perform the new task *and* return the environment to an initial state, effectively performing its own environment reset. Our empirical results demonstrate that incorporating prior data into robotic reinforcement learning enables autonomous learning, substantially improves sample-efficiency of learning, and enables better generalization. Project website: https://sites.google.com/view/ariel-berkeley

**Keywords:** autonomous RL, offline RL, reset-free manipulation

## 1 Introduction

Reinforcement learning (RL) provides a general learning-based framework that, in principle, could be utilized to acquire any goal-directed behavior. However, in practice, the standard RL formulation overlooks many of the challenges that arise in real-world robotic learning. RL problems are classically (though not exclusively) situated in settings where ample exploration can be performed from scratch, the environment can be reset episodically, and the focus is more on attaining the highest possible performance rather than good generalization, such as in the case of playing a video game. Real-world robotic learning problems have very different constraints. With real-world robots, online interaction and exploration are often at a premium, the robot must figure out how to reset the environment itself between attempts, and the natural variability and uncertainty of the real world makes generalization far more essential than squeezing out every bit of final performance.

Humans and animals handle each of these challenges in the real world. However, in contrast to standard robotic RL settings, humans do not approach each new problem with a blank slate: we leverage prior experience to help us acquire new skills quickly, scaffold the process of learning that skill (e.g., by using our existing skills to practice and try again), and utilize representations learned from prior experience to ensure that the new skill is represented in a robust and generalizable way. We study an analogous approach in this work and propose a complete system for extracting useful skills from prior data and applying them to learn new tasks autonomously. Although the individual components of our system are based on previously proposed principles and methods for offline RL [1], multi-task learning with task embeddings [2, 3], and reset-free learning with forward-backward controllers [4, 5, 6, 7, 8], our system combines these components into a novel framework that effectively enables real-world robotic learning and leverages prior data for all of these parts.

Our system consists of two phases: offline learning using the prior data, and mostly autonomous online fine-tuning on a new task (see Figure 1), where only occasional resets are provided. In the offline phase, we use offline RL to extract skills and representations from the prior data. Then, in the online phase, we adapt the learned skills to the new task, alternating between attempting the task and resetting the environment. If the behaviors required to perform the new task and reset the

6th Conference on Robot Learning (CoRL 2022), Auckland, New Zealand.

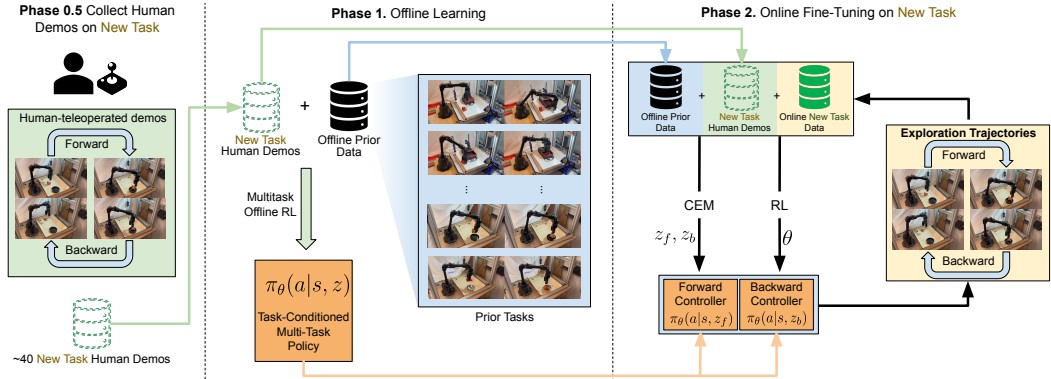

Figure 1: **Overview of our system.** In the offline phase (middle), we train a multi-task policy that captures prior knowledge from an offline dataset of previously experienced tasks and a handful of human-teleoperated demos of a new task (left). Then in the online phase (right) the multi-task policy is used to initialize both a forward policy and a backward (reset) skill for learning the new task, improving learning speed and generalization.

environment are structurally similar to those in the prior data, the skills learned offline will succeed with non-negligible probability, providing a learning signal that allows the agent to adapt its behavior with only a small amount of online data. Moreover, where polices learned from narrow data may overfit to unimportant details of the environment and fail when those details change, polices learned from diverse multi-task data can generalize to new conditions with little to no adaptation.

Our main contribution is demonstrating that incorporating prior data into a reinforcement learning system simultaneously addresses several key challenges in real-world robotic RL: sample-efficiency, zero-shot generalization, and autonomous non-episodic learning. We validate our approach on real-world robotic manipulation, where we show that our method makes it possible to learn new manipulation skills via RL in settings where prior approaches struggle to make progress, reach final performance that is comparable to what the algorithm can attain when provided with "oracle" demonstration data for the new task, and acquire policies that generalize more broadly than those trained only on single-task data.

## 2 Related Work

Prior work has explored learning behaviors from offline datasets, leveraging multi-task data, and reinforcement learning with minimal resets. We build on techniques from each area to devise a complete system for automated robot learning. We briefly review some related works below.

**Accelerating online RL with offline data.** We use an offline RL method [9, 10, 11, 12, 13, 14, 15, 16, 1, 17, 18, 19] to extract useful skills and representations from prior data, but our system includes additional components that allow us to use multi-task offline data and autonomously fine-tune the policy online by learning to both perform the task and reset.

**Multi-task RL.** Similar to methods for multi-task RL [20, 21, 22, 23, 24, 25, 26, 27, 28, 2, 29, 3], we adopt the strategy learning a single policy conditional on a space of tasks. However unlike multi-task RL methods, our focus is adapting this multi-task policy to unseen tasks. Perhaps the closest prior system to ours is MT-Opt [2], which addresses multi-task learning for a similar class of pick-and-place robotic manipulation tasks. However, in contrast to this prior work, our focus is not on how to train a multi-task policy, but specifically on how prior offline data from varied tasks can be leveraged to improve the autonomy and generalization when learning a new task. While Kalashnikov et al. [2] also evaluates fine-tuning to new tasks, our system goes significantly further, aiming to automate resets (which the prior work does not address, instead using an instrumented bin setup), and evaluating the benefits in terms of generalization and efficiency for the new task.

**Reset-free RL.** Our system aims to utilize prior data to autonomously learn robotic skills, leveraging prior experience to both accelerate the learning of a new behavior and the process of learning how to *reset* between attempts. Prior work has tackled this "reset-free" learning problem [30] in a number of ways. Some prior work uses a curriculum-based approach, which relies on the observation that when learning several tasks simultaneously, some tasks reset others, thus forming a curriculum

[31, 32, 33, 4, 34]. While Gupta et al. [31] also uses a multi-task setup to automate resets, this prior paper involves a small number of tasks that all focus on enabling reset-free learning for a particular (single) skill, whereas our focus is specifically on using prior data for *other* tasks to enable a *new* task to be learned as autonomously as possible. Other approaches learn separate controllers for performing the task and resetting the environment [4, 5, 6, 7, 8]. Though our method also learns separate controllers, we initialize them with potentially useful skills extracted from prior data. Most similar to our work, Sharma et al. [8] use prior data in the form of demonstrations to accelerate learning forward and reset behaviors. Our system can use expert demonstrations of the target task, but our main focus is using potentially sub-optimal data from other tasks.

## 3 Autonomous Robotic Reinforcement Learning with Prior Data

First, we formalize our problem statement and describe our evaluation setup. Our goal is to utilize data from previous tasks to address three key challenges in robotic reinforcement learning: sample-efficiency, autonomous learning, and generalization and robustness to unseen test conditions. We assume access to a prior dataset $\mathcal{D}_{\mathrm{prior}}$ of experience collected from multiple training tasks: $\mathcal{D}_{\mathrm{prior}} = \cup_{j=1}^{N} \mathcal{D}_j$ where $\mathcal{D}_j = \{(\mathbf{s}_i^j, \mathbf{a}_i^j, \mathbf{s}_i'^j, r_i^j)\}_{i=1}^{K}$ denotes the chunk of the prior data corresponding to task $\tau_j$. Our goal will be to extract a rich repertoire of skills from this diverse, multi-task prior dataset $\mathcal{D}_{\mathrm{prior}}$ to quickly solve a new target task $\tau^*$. For the prior data to be useful, the training tasks must share structural similarity with the new task. We evaluate two settings, based on the level of similarity between the tasks in $\mathcal{D}_{\mathrm{prior}}$ and $\tau^*$. In the first setting, which we call **direct transfer**, $\tau^*$ is not seen at all during training. For direct transfer to succeed, the training tasks must share enough similarity with the target task that a policy trained on $\mathcal{D}_{\mathrm{prior}}$ achieves a non-zero success rate on $\tau^*$. If this condition does not hold, we evaluate in a second setting which we call **transfer with demonstrations**. We provide additional guidance by augmenting $\mathcal{D}_{\mathrm{prior}}$ with a small number of demonstrations of the target task $\mathcal{D}_{\tau^*}$ (collected by human teleoperation) so that $\mathcal{D}_{\mathrm{prior}} = \mathcal{D}_{\mathrm{prior}} \bigcup \mathcal{D}_{\tau^*}$. (This is labeled as Stage 0.5 in Fig. 1.) In both settings, when the robot is practicing on the target task $\tau^*$, it is provided with minimal external resets, in accordance with the autonomous RL paradigm [30]. The learned agent is then evaluated on the same task $\tau^*$, but this time the environment is reset after every episode.

Our method, which we call Autonomous RobotIc REinforcement Learning (ARIEL), consists of two phases: offline learning using the prior data followed by mostly autonomous online adaptation to a target task (see Figure 1).

### 3.1 Learning From Prior Data with Offline RL

Given a dataset from previous tasks, we first aim to extract skills and representations that could be useful for learning downstream tasks. The multi-task dataset $\mathcal{D}_{\mathrm{prior}}$ can come from many different sources: human demonstrations, sub-optimal data from previous reinforcement learning experiments, or even data collected using (imperfect) hand-engineered policies. To make use of this multi-task dataset for downstream tasks, we train a conditional policy $\pi_\theta(\mathbf{a}|\mathbf{s}, \mathbf{z})$ using offline RL; we utilize the AWAC [1] algorithm in our experiments. This policy is conditioned on a task index $\mathbf{z}$ (represented as a one-hot vector) that informs the agent which task it is performing. Training a single multi-task policy offers several advantages over training separate policies on the prior dataset. First, we are able to learn a representation of policy inputs and outputs (in this case, images and desired change in end-effector pose, respectively) which is shared across tasks, and is therefore more likely to generalize well when faced with new inputs: both through interpolation (if the new task is similar to previously seen tasks), and extrapolation (for novel tasks). Second, when attempting a new task, we only need to sample actions from a single pre-trained policy, as opposed to choosing from an ensemble of pre-trained policies. Finally, this also allows our method to gracefully scale with the number of tasks in the prior dataset. After the offline learning phase, we have a multi-task policy $\pi_\theta(\mathbf{a}|\mathbf{s}, \mathbf{z})$ that can perform each of the tasks in the prior data. In Section 4 we discuss the specific tasks we use in our prior data and how we generate the prior dataset for our experiments.

### 3.2 Autonomous Online Training

After extracting skills from the offline data, our systems adapts the skills to efficiently learn a new task with minimal externally provided environment resets. Since the environment is not frequently reset, the agent must simultaneously learn to perform the task and reset the environment to the initial state so it can continue practicing. Prior work has tackled this problem by learning distinct forward and backward controllers, alternating between applying the forward controller to perform the task and applying the backward controller to reset the task [4, 5, 6, 7, 8]. We adopt an analogous strategy,

but we leverage the multi-task policy described in the previous section to bootstrap learning in both the forward and backward direction, which both helps to overcome the exploration challenge and provide automated resets more quickly than if the backward controller were trained from scratch.

To enable this, we fine-tune the parameters of the policy $\pi_\theta(\mathbf{a}|\mathbf{s}, \mathbf{z})$ obtained after the offline phase with online data from the new task $\tau^*$, which we denote as $\mathcal{D}_{\tau^*}$, similarly to prior work that uses online fine-tuning to improve upon policies learned via offline RL alone [1]. However, since we have a multi-task policy, we must confront the question of which skill(s), parameterized by task indices $\mathbf{z}$, should be chosen to start fine-tuning for the forward and backward controller. If demonstrations of the target task were provided (in the transfer with demonstrations setting), we can use the task indices associated with these demonstrations. Otherwise, if no demonstrations were provided (the direct transfer setting), we must automatically determine which prior skill (or combination of prior skills) to fine-tune. To do this, we treat the task index $\mathbf{z}$ as a continuous, learnable parameter and use data collected during online reinforcement learning to select the $\mathbf{z}$'s best suited for the given new task. In this sense, we treat $\mathbf{z}$ as a task embedding [3] that is automatically adapted for the task at hand, similar to prior work in meta-reinforcement learning [35].

Since we aim to learn both a forward and backward controller, we optimize for two skills, parameterized by two distinct task embeddings $\mathbf{z}_f$ and $\mathbf{z}_b$, that respectively perform the task and reset the environment. Conditioning the policy on $\mathbf{z}_f$ results in the forward controller $\pi_\theta(\mathbf{a}|\mathbf{s}, \mathbf{z}_f)$ that performs the task, whereas conditioning the policy on $\mathbf{z}_b$ results in the backward controller $\pi_\theta(\mathbf{a}|\mathbf{s}, \mathbf{z}_b)$ that resets the environment according to the initial state distribution of the task $\tau^*$. During fine-tuning we alternate between applying the forward controller and backward controller, optimizing the forward controller with rewards for completing the task $r_f(\mathbf{s}, \mathbf{a})$ and the backward controller with rewards for returning to the initial state distribution $r_b(\mathbf{s}, \mathbf{a})$. We optimize the parameters $\mathbf{z}_f$ and $\mathbf{z}_r$ in tandem with the policy and value function parameters $\theta$ and $\phi$. The parameters of the policy and value function are updated using using an actor-critic algorithm (we use AWAC [1]). The embeddings $\mathbf{z}_f$ and $\mathbf{z}_r$ are optimized differently, as we discuss next.

---

**Algorithm 1** ARIEL online adaptation

**Input:** pre-trained policy $\pi_\theta(\mathbf{a}|\mathbf{s}, \mathbf{z})$, Q-function $Q_\phi^\pi(\mathbf{s}, \mathbf{a}, \mathbf{z})$

1: Initialize buffer $\mathcal{B}$, CEM models $q_f, q_b$
2: Fit $q_f(\mathbf{z})$ and $q_b(\mathbf{z})$ to offline task indices
3: $d \leftarrow f$ // task direction, f is forward, b is backward
4: **while not** done **do**
5:    $s \sim \rho(\mathbf{s}_0)$ // sample initial state (i.e external reset)
6:    **for** $N$ steps **do**
7:       // sample new task embedding at max trajectory length
8:       **if** steps % $M$ == 0 **then**
9:          $\mathbf{z} \sim q_d$
10:       **end if**
11:       Sample $\mathbf{a} \sim \pi(\cdot|\mathbf{s}, \mathbf{z})$, observe $\mathbf{s}' \sim p(\cdot|\mathbf{s}, \mathbf{a})$
12:       $\mathcal{B} \leftarrow \mathcal{B} \cup \{(\mathbf{s}, \mathbf{a}, r_d(\mathbf{s}', \mathbf{a}), \mathbf{s}', \mathbf{z}\}$
13:       **if** $r_d(\mathbf{s}', \mathbf{a})$ **then**
14:          `switch(`$d$`)` // switch task direction
15:       **end if**
16:       update $\pi_\theta$, $Q_\phi^\pi$ w/ RL, update $q_f(\mathbf{z})$, $q_b(\mathbf{z})$ w/ CEM
17:       $\mathbf{s} \leftarrow \mathbf{s}'$
18:    **end for**
19: **end while**

---

**Optimizing task embeddings $\mathbf{z}_f$ and $\mathbf{z}_b$.** Since our goal is to learn the target task $\tau^*$ as quickly as possible, and with minimal resets, we need to effectively explore the environment during online training. An ideal approach that explores effectively must not prematurely commit to a particular value of $\mathbf{z}_f$ and $\mathbf{z}_b$ until the online experience clearly indicates so, and at the same time, it should be fast at collapsing to the correct values of $\mathbf{z}_f$ and $\mathbf{z}_b$, when it is apparent from the online experience gathered until then. We found that the cross-entropy method (CEM) [36] provided us with a favorable balance between exploration and exploitation. CEM provides a way to initialize the task embedding to a *distribution*, which ensures that we do not have to commit to fine-tuning any one particular skill. Since we want to optimize both the forward and backward embeddings $\mathbf{z}_f$ and $\mathbf{z}_b$, we fit two sampling distributions with CEM, $q_f(\mathbf{z})$ and $q_b(\mathbf{z})$. The distribution of tasks in the prior data $\mathcal{D}_{\text{prior}}$ provides us with a natural (and hopefully informative) prior to initialize $q_f(\mathbf{z})$ and $q_b(\mathbf{z})$. On each iteration of training, we sample a $\mathbf{z}$ from either $q_f(\mathbf{z})$ or $q_b(\mathbf{z})$ and roll out the policy conditioned on this $\mathbf{z}$ value for $M$ steps. These $M$-step trajectories essentially correspond to an episode in episodic RL, however the environment is (typically) not reset afterward. We alternate between sampling the $\mathbf{z}$ from $q_f(\mathbf{z})$ or $q_b(\mathbf{z})$ to alternate between applying the forward and reset controllers. The reward function correspondingly alternates between $r_f(\mathbf{s}, \mathbf{a})$ and $r_b(\mathbf{s}, \mathbf{a})$. We periodically refit $q_f(\mathbf{z})$ and $q_b(\mathbf{z})$ to the $J$ most recently sampled $\mathbf{z}$'s that resulted in trajectories that successfully completed or reset the task, respectively. We update the policy and value function parameters $\theta$ and $\phi$ using the update steps from an RL algorithm as summarized in Algorithm 1. Every $N$ steps, we provide an external reset to the environment. However, $N >> M$, so the external resets are infrequent. For example, in one of our real world tasks, $M = 20$, and $N = 400$. In theory, we could run our method without any manual resets at all, but we

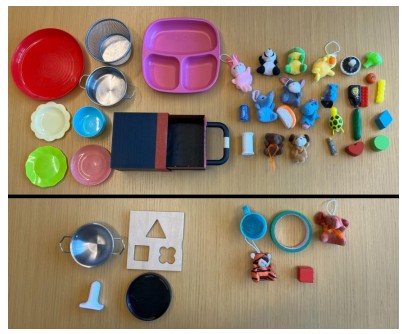

(a) Objects and Containers.

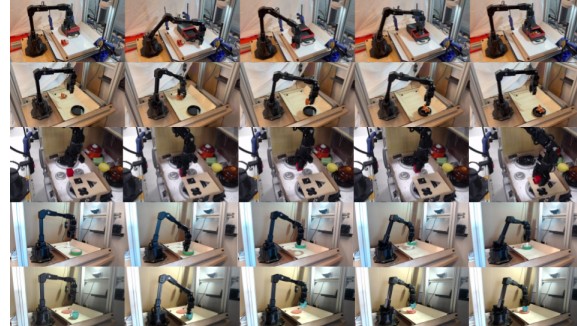

(b) Downstream Tasks.

Figure 2: **(a):** Objects used to construct the train tasks (upper) and test tasks (lower) in our experiments. **(b):** Selected test tasks on our real-world robotic setup.

found a small number of manual resets necessary to recover from certain states, similarly to prior work on reset-free RL [5].

### 3.3 Algorithm Summary

To summarize, we first use offline RL on $\mathcal{D}_{\text{prior}}$ to obtain a multi-task policy $\pi_\theta(\mathbf{a}|\mathbf{s}, \mathbf{z})$ and value function $Q_\phi(\mathbf{s}, \mathbf{a}, \mathbf{z})$. Then, in the online phase (see Algorithm 1), we adapt this policy and value function to a target task. Notably, the offline phase occurs only once, but the online phase can be run many times to efficiently learn many new tasks. During online adaptation, we use $\pi_\theta(\mathbf{a}|\mathbf{s}, \mathbf{z})$ to initialize a forward and backward controller that perform and and reset the task respectively. Alternating between applying the forward and backward controller allows the agent to autonomously practice the target task. Our system enables us to learn the new task efficiently and with minimal environment resets by leveraging prior data from previous tasks. We provide further implementation details in the Appendix.

## 4 Experiment Setup

We now describe our experimental setup for data collection, training, and evaluation.

**Robotic platform.** We evaluate our approach using a 6-DoF WidowX robotic arm (see Figure 2b). The 7D action space of the robot consists of 6D Cartesian end-effector motion, corresponding to relative changes in pose, as well as one dimension to control the opening and closing of the parallel jaw gripper. The state observations consist of the joint angles and RGB images from an overhead camera, which are 48×48 in simulation and either 64×64 or 128×128 in the real world.

**Evaluation tasks.** To effectively leverage data from prior tasks to solve new tasks, the prior tasks must share structural similarity with the new tasks. In our evaluation, we study a variety of pick and place tasks, where the shared structure is that each task involves picking up an object and placing it into or on top of some container or object, and the variability between the tasks corresponds to different containers or placement targets, different objects to be picked, and potentially even additional manipulations that must be performed on the container (e.g., opening a drawer prior to putting something inside). The tasks are performed in one of two scenes: a simpler scene where the objects and containers are in a tray, and a more visually complex kitchen scene. Both scenes contain distractor objects that are not involved in the tasks, and the robot is required to correctly learn which object to interact with, how to pick it, and where to place it. All tasks have sparse rewards evaluated using a simple hand-designed vision system, with a reward of +1 if the robot completes the forward (or backward) task, and 0 otherwise. While such rewards are easy to define, they present a substantial challenge for RL.

The test tasks that we fine-tune on, some of which are shown in Fig. 2b, span a range of difficulties and consist of picking and placing with objects and containers that were not seen in the prior data. Three of the tasks are in-distribution and hence similar to the prior data: placing a toy tiger in a drawer (which requires also opening the drawer, then closing it again to reset), placing a toy elephant in a pot, and placing a toy tiger on a lid. The other four tasks are out-of-distribution: placing a cup on a saucer, inserting a ring onto a peg, sorting a block into its correct slot, and moving a pot to the stove. The cup, ring, and pot require significantly different grasping behaviors than the objects seen

in the prior data since they must be grasped from the edge. Additionally, successfully placing the ring on the peg and the block in the slot requires greater precision than any of the prior tasks.

**Generating prior data.** Our system can use prior data from any reasonable source, including demonstrations and prior RL runs, but for simplicity and consistency we employ a simple scripted policy to collect to collect data for placing a wide variety of objects into containers (see Fig. 2a). This provides a largely autonomous strategy to collect mediocre data for many objects. While this scripted policy fails very frequently, offline RL can make use of such mediocre data effectively to pretrain value functions and policies for downstream fine-tuning. We provide further details of the data we generated in the Appendix.

**Providing demonstrations.** In the transfer with demonstrations setting, we augment the scripted data with around 40 demonstrations of the target task collected by teleoperation with a 3DConnexion SpaceMouse Compact. Rewards are specified by a query at the end of each trajectory. This system allows specification of new tasks with no additional engineering effort.

**Online training setup.** Online training includes infrequent resets, to address the case where an object becomes stuck in a hard-to-reach (or unreachable) part of the workspace, but otherwise requires the robot to train autonomously. The resets are provided every 80 trials in simulation (about every hour), and roughly every 20-30 trials (about every 20-30 minutes) in the real world.

## 5 Experimental Results

Our experiments aim to evaluate the hypothesis that leveraging prior data from other tasks can simultaneously address multiple challenges in autonomous real-world robotic RL. To this end, we evaluate ARIEL on seven real-world downstream tasks and a suite of simulated robotic manipulation tasks. Our goal is to answer the following questions: **(1)** Does leveraging prior data enable mostly autonomous learning of both in-distribution and out-of-distribution tasks in the real-world? **(2)** Does ARIEL produce policies that better generalize to new conditions? **(3)** How does ARIEL compare to prior methods for learning with minimal resets that do not use prior data? **(4)** Does leveraging prior data via ARIEL lead to gains in sample-efficiency for a new task? Videos of the experiments can be found on the project website: https://sites.google.com/view/ariel-berkeley

### 5.1 Evaluating ARIEL in the Real World

We first evaluate the performance of ARIEL when training online to solve new instances of the pick and place tasks in the real world. After pretraining on the datasets discussed in Section 4, we evaluate ARIEL on both in-distribution and out-of-distribution tasks using the direct transfer and transfer with demonstrations settings, respectively. We evaluate success rates of the policy learned in the offline phase, the policy learned after 100 trials (1.5 hours) of online training, and the final policy after 600 trials (9 hours).

**Autonomous fine-tuning with infrequent resets with ARIEL significantly improves real-world performance.** Table 1 shows the success rate of the forward controller for both the in-distribution and out-of-distribution tasks increases moderately after 100 trials (e.g., from 2/10 to 4/10 on *put tiger in drawer*) and substantially by the end of online training (e.g., from 4/10 to **9/10**). This indicates that ARIEL successfully optimizes the embeddings and policy parameters, with minimal resets, to perform the new task. Similar results for the performance of the backward controller are included in the Appendix. Notably, fine-tuning from only the small set of demonstrations (for the out-of-distribution tasks) largely fails, meaning that the agent is successfully transferring skills from the other tasks in the prior data. Additionally, the autonomous fine-tuning phase does not simply robustify the previously seen behaviors, but can also lead to the emergence of new behaviors. In the *insert ring onto peg* task, the fine-tuned reset policy also learned useful retrying behaviors, such as shifting the ring to a more favorable grasping position so that the peg would not obstruct the robot from taking off the ring, a behavior it did not exhibit from offline training alone. This is best seen in the accompanying video.

**Training on diverse prior data improves zero-shot generalization in the real world.** Next, we investigate whether initializing from prior data provides ARIEL with more robust representations that generalize more effectively to new conditions without any further adaptation. To study this hypothesis, we additionally evaluate the fine-tuned policies for the *tiger on lid*, *ring on peg*, and *cup on saucer* tasks in a setting where we swap out the object (but keep the same container). We do not expect the policies to succeed consistently in this case, but we see in Table 2 that ARIEL policies exhibit some generalization, attaining an average success rate of **54%** at the best epoch. A baseline

|  | Task | Offline Only | 100 Trials | 600 Trials | Only Demos |
|---|---|---|---|---|---|
| **Direct Transfer** | Put Tiger in Drawer | 2/10 | 4/10 | **9/10** | N/A |
|  | Put Elephant in Pot | 1/10 | 2/10 | **7/10** | N/A |
|  | Put Tiger on Lid | 0/10 | 4/10 | **7/10** | N/A |
| **Transfer with Demos** | Put Cup on Saucer | 7/10 | **9/10** | - | 0/10 |
|  | Insert Ring onto Peg | 4/10 | **10/10** | - | 0/10 |
|  | Put Block in Slot | 5/10 | **8/10** | - | 0/10 |
|  | Move Pot to Stove | 2/10 | **9/10** | - | 2/10 |

Table 1: **Real-world evaluation of the forward controller.** We evaluate the success rates of the forward controller over the course of online fine-tuning. The first 3 tasks are in-distribution and use the direct transfer setting. The next 4 tasks are out-of-distribution and use the transfer with demonstrations setting. For tasks in the transfer with demonstrations setting, we also evaluate a policy trained only the demonstrations. Success rates are calculated using 10 trials where each trial starts with different initial object positions.

| Target Task | Unseen Task | ARIEL by # Trials | | | Target Data Only |
|---|---|---|---|---|---|
|  |  | 100 | 360 | 600 | Best Epoch |
| Put Tiger on Lid | Put Monkey on Lid | 5/10 | **6/10** | 5/10 | 6/10 |
| Put Tiger on Lid | Put Rabbit on Lid | 2/10 | **4/10** | 0/10 | 0/10 |
| Put Tiger on Lid | Put Hippo on Lid | 3/10 | **5/10** | 0/10 | 0/10 |
| Put Cup on Saucer | Put Orange Pot on Saucer | **6/10** | - | - | 0/10 |
| Insert Ring on Peg | Insert Black Ring on Peg | **6/10** | - | - | 0/10 |

Table 2: **Testing zero-shot generalization.** We evaluate the zero-shot generalization performance of policies fine-tuned with ARIEL. We take the policies obtained at different stages of online fine-tuning on the *tiger on lid*, *cup on saucer*, and *ring on peg* target tasks and evaluate them zero-shot on unseen tasks with interchanged objects. We compare with a baseline trained on only the data collected for the target tasks, without first training on the prior data (Target Data Only) and find that policies initialized with diverse prior data (ARIEL) generalize better to unseen tasks. We show success rates calculated using 10 trials where each trial starts with different initial object positions.

that uses only the data collected for each target task (i.e., either only *tiger on lid*, *ring on peg*, or *cup on saucer*) only succeeds in 12% of trials on the new objects. This suggests that pretraining on multi-task prior data does effectively boost generalization.

Interestingly, we observe that while fine-tuning improves the performance of ARIEL policies on the target tasks (see Table 1), fine-tuning on the target tasks adversely affects the zero-shot generalization performance on a new object. This is to be expected, since even though our online training procedure replays the prior data, we train with an increasing proportion of online data as online training goes on, making the policy more specialized to the target task. This observation is also consistent with prior RL works [37, 38] that note a drop in performance on test tasks as more updates are made on the training tasks. We anticipate that more principled early stopping or replay of the prior data could alleviate this issue, which is an interesting avenue for future work.

## 5.2 Evaluating ARIEL in Simulation

Lastly, we briefly present simulated experiments to compare ARIEL to prior autonomous learning approaches and ablations. Further details and analysis can be found in the Appendix.

**ARIEL outperforms prior approaches for autonomous learning.** We compare to **R3L** [6], which alternates between training a forward controller to optimize a task-completion reward and training a perturbation controller that optimizes a novelty exploration bonus. This method has been proposed specifically to handle reset-free learning for real-world RL problems. We also compare to an ablation (**Multi-task RL**) that does not optimize the task embeddings $z_f$ and $z_b$ over the course of autonomous online fine-tuning, but instead fine-tunes the policy using a fixed task index (that was unused during pre-training). Finally, we compare to an "oracle" version of our approach, labeled **ARIEL + resets**, which assumes access to external resets at the end of each episode. Figure 3 shows that ARIEL and its oracle reset variant are the only methods that succeed at learning the new tasks, and the full ARIEL method closely matches the oracle.

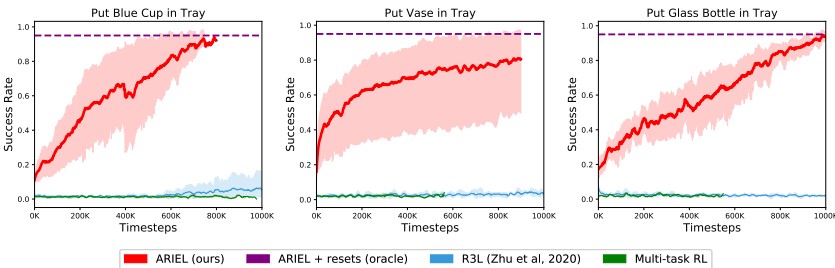

Figure 3: **Limited Resets.** Simulated experiments comparing our method to prior works in reset-free and multi-task learning. Ours is the only method that learns in this limited-reset setting. Results are averaged over 4 random seeds.

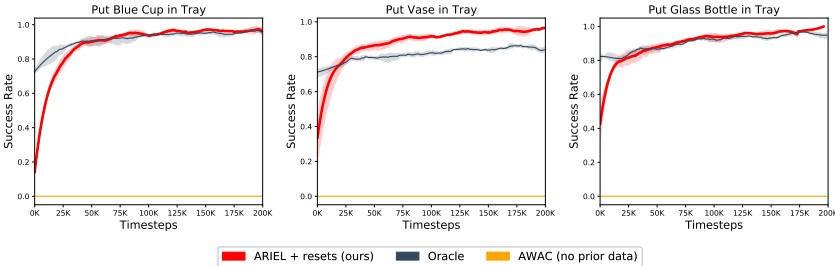

Figure 4: **Sample Efficiency.** Simulated experiments comparing our method to (1) an oracle with offline data for the downstream task, and (2) learning without any prior data. Unlike in Figure 3, these experiments had resets at the end of every episode. Our method adapts almost as quickly as the oracle approach, whereas RL from scratch fails to make any progress. Results are averaged over 4 random seeds.

**ARIEL enables sample-efficient learning of the new task.** Next, we compare our method to simply applying RL to the target task, using the same RL algorithm as our system (**AWAC (no prior data)**), as well as an **oracle** method which is trained offline on data coming from only the single new task of interest and then trained online on the same task. Perhaps unsurprisingly, we found that learning from scratch fails in the challenging setting of learning from images and sparse rewards. However, our method learns just as efficiently as the oracle despite only having access to offline data for other tasks and not the new task of interest. Thus, our method demonstrates that properly leveraging prior data can improve sample-efficiency even when data is only available for other tasks.

## 6 Discussion and Limitations

In this work, we proposed overcoming the numerous challenges of real-world robotic reinforcement learning through better leveraging prior datasets. While reinforcement learning often involves acquiring a skill from scratch, such a formalism is not well-suited to real world conditions where running online data collection is expensive, and previously collected datasets are often accessible. While the use of previously collected datasets for tackling a particular challenge (such as that of sample efficiency [1], or better generalization [39]) has been addressed in prior work, our work aims to address such challenges jointly in a single robotic learning system, which leverages prior data as the primary mechanism to facilitate autonomy during online training, improve efficiency, and boost generalization. In our experiments, we show that our method is able to make consistent progress in online fine-tuning despite having to learn from sparse rewards, matching or exceeding the final performance of an oracle baseline that receives prior data *of the actual test task*, and significantly outperforming a baseline without prior data. This supports our central hypothesis that prior data can be a significant facilitator for real-world robotic RL.

**Limitations.** The primary limitation of our approach is that our online target tasks are structurally quite similar to the prior data used for offline pretraining (particularly in the direct transfer setting). While the objects seen during online training are different, they do not require significantly distinct physical skills. We believe that increasing the diversity and breadth of the prior data will make the offline pretrained model useful for a broader range of new tasks. Exploring this direction by scaling up the proposed approach is an exciting avenue for future work.

**Acknowledgments**

This research was supported by the Office of Naval Research, NSF CAREER IIS-1651843, and ARO W911NF-21-1-0097, with computing support from Google.

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
