# OpenReview forum: "Don’t Start From Scratch: Leveraging Prior Data to Automate Robotic Reinforcement Learning"
_robot-learning.org/CoRL/2022/Conference — CoRL 2022 Poster_

### Official Review · Reviewer_YCu7 · 2022-07-26

**Originality:** Very Good
**Technical Quality:** Good
**Clarity Of Presentation:** Good
**Impact:** 4

**Recommendation:**

Weak Accept: I recommend accepting the paper, but will not argue for my recommendation if the majority of other reviewers have a different opinion.

**Summary:**

Pre-training with large offline data and fine-tuning on a small amount of online data is a sample-efficient way of learning a new task. However, most recent approaches require explicit environment resetting for online data collection. This work proposes to minimize manual resetting of the environment when learning a new task in the real world leveraging offline multi-task data. First, from the offline multi-task data, the proposed method, ARIEL, learns a multi-task policy conditioned on a task embedding $\mathbf{z}$ using offline RL. Then, during the online fine-tuning phase, ARIEL searches for a task embedding for the forward task as well as another task embedding for the backward task (environment reset). ARIEL iterates the forward and backward tasks, which ideally resets the environment and performs the target task without human intervention. The real-world experiments show that ARIEL achieves more than 70% success rates with only 5 to 30 manual resets.

**Issues:**

Please refer to the above weaknesses and questions.

**Quality Of The Limitations Section:**

Additional details required

**Reviewer Expertise:**

3: The reviewer is fairly confident that the evaluation is correct

**Robotics Focus:**

Sufficient demonstration on hardware

**Strengths And Weaknesses:**

### Strenghts

* The idea of utilizing a multi-task policy for both forward and backward (reset) policies makes sense and the paper empirically shows its effectiveness.

* The real-world experiments are exhaustively conducted with many different objects and in three different environment setups.

### Weaknesses

* In RL, covering diverse initial states is important for robustness and generalization of the learned policy. However, without explicit randomized resetting, the proposed backward controller may learn to reset to a few (easy to reset) initial states, which maximizes its own reward. It would be great if the authors discuss this potential issue.

* The proposed alternating scheme is appealing in automating online reinforcement learning with less human intervention. However, one of the critical assumptions in this work is that both forward and backward controllers can be achieved with a few random shootings. The reviewer is wondering whether this is a practical assumption for robot learning.

* To show the sufficient difference between training tasks and testing tasks, another interesting point could be freezing the policy after pre-training and only finding appropriate $\mathbf{z}$ using CEM. If this works as well as the proposed method, it could mean the prior data is too similar to the testing tasks, which means the experimental setup might be too easy.

* The proposed method assumes a task embedding corresponds to a task. However, in many cases, the offline data trajectories can be compositional (combination of multiple behaviors) and the target task may require stitching fo multiple of these trajectories. The proposed method does not seem trivial to scale to this common scenario.

### Questions and suggestions

* Figure 2b is too small, compact, and not descriptive without enough explanation. It would be better to include only two examples (one from in-distribution and another one from out-of-distribution tasks) and provide sufficient descriptions.

* L242-243: placing the ring on the peg does not seem to require "greater precision" than the prior tasks.

* In L244-245, while the paper claims that the proposed method can utilize data from any reasonable source, the proposed method requires reward annotation, which is not available in reward-free demonstrations (e.g. play data, multi-task expert demonstrations). Given that the proposed method can potentially work with sparsely-labeled data using offline RL, it may not be a big issue, but it should be clarified.

* In L253, what does it mean that rewards are specified at the end of the trajectory?. The reviewer thinks rewards for the target demonstrations should be all 1 at the end. Or, does it mean that the reward is labeled depending on the forward or backward tasks?

* What is the definition of "trial" in Section 5 and 6? Does this mean $M$ steps or $2M$ steps?

* In Table 1, why do in-distribution tasks perform worse than out-of-distribution tasks?

* In Table 1, including a baseline with episodic reset as an upper bound can be helpful to understand how much "reset" information is important for learning these tasks.

* In Table 1 and Table 2, which experiment uses 40 downstream task demonstrations? This is not clear.

* The explanation in L291-L293 about the experiments for Table 2 is a bit confusing. Are the testing tasks unseen during online fine-tuning? More specifically, "Put X in Container" tasks are evaluated using a policy pre-trained from multi-task prior data and fine-tuned for "Put Tiger on Lid"?

* In Table 2, how many trials were used for training "Single Task Data Only"?

* In A.2.1, there are three different environments, "Tray Container", "Tray Drawer", and "Kitchen". For offline training of ARIEL, are data from all three environments used together? or data from the same environment with the target task is used for offline training?

* In Algorithm 1, "steps" should be reset to 0 and new $\mathbf{z}$ should be sampled, like L9, when the task direction $d$ is reversed in L14.

* In Algorithm 1, L2, what does it mean by "fit $q_f(\mathbf{z})$ and $q_b(\mathbf{z})$ to offline task indices?

* In Algorithm 1, L5, $d$ should be also reset to $f$ as the environment state is reset. Simply moving L3 to L5 can work.

* How much randomness in object poses and robot poses for each trial of evaluation in Table 1 and 2?

* In Table 1, how is "Offline Only" evaluated? More specifically, which $\mathbf{z}$ is used for this experiment? And, how can out-of-distribution tasks have such high success rates only with offline only?

* A bit more detailed description of the task indices $\mathbf{z}$ can be helpful to understand the proposed method. The paper mentions that $\mathbf{z}$ is a task embedding from Hausman et al. in L159, but more details about how it is trained can be elaborated in the paper.

* It could be interesting to see how success rates of the forward and backward controllers over the online training.

* Many references are from arXiv. Please cite their conference or journal versions.


**Summary Of Recommendation:**

This paper proposes a (semi) autonomous reinforcement learning leveraging offline multi-task data, and the empirical results demonstrate its effectiveness and autonomy (less than 30 manual resets required). Although the experimental tasks look a bit simple and the presentation should be improved, the proposed idea is novel and the results show some potential for autonomous RL.


**After rebuttal**

Thank the authors for the response. The rebuttal clarified all my confusion, but the weaknesses are still there and a more in-depth discussion about the reviewers' concerns would make the paper stronger. Overall, I believe the proposed method shows an interesting research direction and empirically demonstrated its potential in real-world scenarios; thus, I would like to accept this paper.

---

> ### Author Response · Authors · 2022-08-18
> **Author Response to Reviewer YCu7**
>
> Thank you for your detailed review. We agree that without explicit randomized resetting, the backward controller may collapse to only resetting to a few easy initial states. This was not an insurmountable obstacle in our experiments, even though objects could start from anywhere in the workspace. We will add discussion of this potential limitation.
>
> Please see our top level response for discussion of our assumption that the forward and backward controllers have an initial non-zero success rate on the new task.
>
> We acknowledge that our method is not suited for problems that require stitching together multiple behaviors from the prior data to solve the task, and we will add discussion of this limitation. Our intention was to show that prior data can be useful for initializing both forward and reset skills. In principle, our method could be extended to plan for the optimal sequence of prior skills to initialize the forward and backward controllers, but this is outside the scope of the present paper.
>
> In regard to our claim that our method can use prior data from “any reasonable source,” it is true that this prior data must be annotated with rewards. But we demonstrate that our method works with sparse rewards, so the reward-labeling burden is not high. When labeling the demonstrations with rewards, the user specifies where in the trajectory the task was achieved and whether the task was in the forward or backward direction. This is what we mean by “rewards are specified at the end of the trajectory.” We will add these clarifications.
>
> To clarify Table 2, your understanding of these experiments is correct. We take the policies from different stages of the “Put Tiger on Lid” experiment in Table 1 and evaluate them on the “Put X in Container” tasks listed in Table 2. We compare to the baseline “Single Task Data Only,” where we train on only the data collected in the corresponding target task in Table 1. These experiments were designed to test whether initially training on diverse, multi-task prior data and then fine-tuning on a single task results in a policy that generalizes better than a policy obtained by only training on a single task. Please see the revised PDF attached to our top level response where we have reformatted Table 2 to make this more clear. Specifically, we added a column to indicate which target task policy is used for each unseen task, and we have renamed the “Single Task Data Only” baseline to “Target Task Data Only.”
>
> While the task embedding $z$ serves a similar role as in Hausman et al., we do not train $z$ with gradient descent but instead use the Cross Entropy Method (CEM) as described in section 4.2. We perform CEM with two sampling distributions, $q_f(z)$ and $q_b(z)$, represented by Gaussian mixture models. To provide a reasonable starting point for CEM, we initially fit the Gaussian mixture models to the set of offline task indices. This way, the search for the optimal task embedding will start near the offline task indices. This is what we mean by "fit $q_f(z)$ and $q_b(z)$ to offline task indices" in Algorithm 1, L2.
>
> For the offline only evaluation in Table 1, $z$ is sampled from the initial CEM sampling distributions. Thus, $z$ is a random sample around the offline tasks indices. As for why the the out-of-distribution tasks have relatively high initial success rates, we use the transfer with demonstrations setting for the OOD tasks, so the offline only numbers reflect training on the prior data in addition to the demonstrations. Additionally, for these tasks we do not have to search for a task embedding $z$, since we can just use the task indices associated with demonstrations from the beginning. This also increases the success rate.
>
> To address some smaller questions:
> - For the “Single Task Data Only” baseline in Table 2, we train on the data collected for the experiments in Table 1, consisting of 100-600 trials.
> - We do not combine the Tray Container, Tray Drawer, and Kitchen datasets. For each target task we only use the dataset in the same environment. We did not expect to obtain much benefit from combining the data due to differences in robot models and camera angles between the datasets.
> - Initial object positions are randomized for each trial in Tables 1 and 2, but the robot always starts from the same neutral pose.
> - The success rates of the backward controller are provided in Table 7 in the Appendix.
> - We use “trial” to refer to either an attempt at the forward or backward direction (i.e M steps).
> - In Table 1, the tasks that use demonstrations are “Put Cup on Saucer,” “Insert Ring onto Peg,” and “Put Block in Correct Slot.” Please see the revised PDF attached to our top level response where we have reformatted Table 1 to make this clear.
> - We will update Figure 2B and the algorithm pseudocode with your comments and update our references with the conference or journal versions.

---

### Official Review · Reviewer_BwFL · 2022-07-27

**Originality:** Good
**Technical Quality:** Good
**Clarity Of Presentation:** Excellent
**Impact:** 3

**Recommendation:**

Weak Accept: I recommend accepting the paper, but will not argue for my recommendation if the majority of other reviewers have a different opinion.

**Summary:**

The paper discusses a system to leverage prior data from related but distinct tasks to increase learning efficiency and generalization in new task. The system works in two steps: 1. Using offline data with consisting of different tasks with task ids to learn a task-conditional policy using offline RL. 2. Optimize task-conditioning embedding during deployment for the new task. To reduce the human supervision involved in resets the method proposes to extract two different embeddings one that accomplishes the tasks and the other that resets the task. The experiments are shown in both real world robot and simulation.

**Issues:**

1. Limited baselines: Can the authors present baselines for their real-world and simulated robot experiments. If the comparisons are not feasible, can the authors describe in the paper what makes such a comparison not possible.
2. Technical novelty: The work seems to borrow heavily from previous works [8,31] (references from paper) on reset free learning and offline multitask learnings. Can the authors comment on the significance of the new proposed methods in comparison to previous work?

**Quality Of The Limitations Section:**

Limitations are addressed clearly

**Reviewer Expertise:**

4: The reviewer is confident but not absolutely certain that the evaluation is correct

**Robotics Focus:**

Sufficient demonstration on hardware

**Strengths And Weaknesses:**

Strengths:

1. The paper presents a relatively novel combination of existing known modules for learning with offline data and learning reset policies.
2. The paper evaluates the results on real world and simulated domains and provide comprehensive results to show the validity of the method. The experiments demonstrate the superior generalization ability of the system arising as a result of learning from multiple tasks and increased learning efficiency.

Weaknesses:

1. Limited baselines: The paper evaluates ARIEL on the real world tasks but does not present a comparison with any baseline. Potential baselines to be compared to are: [2,29,31,32,33](references from paper) and common RL baselines which are more suited to sparse rewards like HER[1,2,3] (references below).
2. The experiments use a short horizon of M=20 which is small for a number of locomotion and navigation tasks. It is not clear how this method would scale with horizon since the start-state distribution for learning the reset policy can grow exponentially.

[1] :Hindsight Experience Replay [Marcin Andrychowicz] et al

[2]: [Replacing rewards with examples: Example-based policy search via recursive classification],Eysenbach et al

[3] [C-Learning: Learning to Achieve Goals via Recursive Classification], Eysenbach et al

**Summary Of Recommendation:**

The paper presents a novel combination of existing modules for autonomous robot learning and demonstrates that near-autonomous learning is possible with similar tasks in the short-horizon manipulation domains. The paper can be strengthened by more extensive comparisons to baselines to properly situate how their work progresses the field.

---

> ### Author Response · Authors · 2022-08-18
> **Author Response to Reviewer BwFL**
>
> Thank you for your thoughtful review. Please see our top level response where we address your concerns regarding novelty. In regard to additional baselines, we compared with what we consider the most relevant methods including R3L [6] and a variant of MT-Opt [2] modified for our setting. We agree that the baselines you suggest are related to our work, and we will add further discussion of them, however they do not address our problem setting:
>
> - Yu et al. [29] does not address fine-tuning policies online.
> - While Gupta et al. [31] does use a multi-task setup to automate resets, they focus on enabling reset-free learning for a particular (single) skill, whereas our focus is specifically on using prior data for other tasks to enable a new task to be learned as autonomously as possible.
> - Similarly, Ha et al. [33] deploy multiple policies in sequence to learn a particular single skill, however our focus is on leveraging data from prior tasks to learn new tasks autonomously.
> - Lu et al. [32] addresses the problem of lifelong learning in non-stationary settings where the underlying MDP changes throughout online training. We fine-tune in a single MDP.
> - Andrychowicz et al. [1] and Eysenbach at al. [3] both address the goal-conditioned RL setting, where hindsight experience replay can help overcome sparse rewards. We do not consider the goal-conditioned setting.
> - Eysenbach et al. [2] does not consider the autonomous/reset-free setting.
>
> In regard to the horizon length in our experiments, we actually use a horizon ranging from 20-35 (full details can be found in Table 5 in the Appendix). We do agree that as the space of possible initial states increases, the tasks become harder for both the forward and backward controllers. However, our method succeeded in our robotic manipulation experiments where objects could end up anywhere in the workspace. We will add further discussion of how the range of possible initial states affects the difficulty of the tasks.

---

### Official Review · Reviewer_rVQq · 2022-08-01

**Originality:** Fair
**Technical Quality:** Very Good
**Clarity Of Presentation:** Good
**Impact:** 3

**Recommendation:**

Weak Accept: I recommend accepting the paper, but will not argue for my recommendation if the majority of other reviewers have a different opinion.

**Summary:**

Real-world robotic reinforcement learning (RL) typically requires time-consuming data collection and frequent human intervention to reset the environment. The offline-RL approach provides a way to utilize previously collected data to bootstrap the learning process so that a new task can be learned with a small number of interactions. In this work, the authors study a problem where such offline data is available from several tasks (along with their task labels). The data may be collected with optimal or sub-optimal policies.  The authors propose a complete system for extracting useful skills from prior data and applying them to learn new tasks autonomously. When faced with a new task, the proposed system adapts previously learned skills to quickly learn to both perform the new task and return the environment to an initial state, effectively performing its own environment reset. The authors show the effectiveness of their system with extensive physical and simulated experiments.

**Issues:**

Maybe the contribution of the paper could have been clearly mentioned instead of "demonstrating already known" facts as a contribution. For instance, we already know using prior data makes RL samples efficient, there is nothing to demonstrate here again in my opinion.

Line 65: "we adopt the strategy learning a single policy conditional on a space of tasks" => we adopt the strategy *of* learning a single policy condition*ed* on a space of tasks.

There are several citations that are now published. But the arxiv versions are still cited on the paper. Please use the published versions. For instance, [31] should be ICRA 2021.

line 246: Typo here "...policy to collect to collect data for...."

**Quality Of The Limitations Section:**

Additional details required

**Reviewer Expertise:**

3: The reviewer is fairly confident that the evaluation is correct

**Robotics Focus:**

Sufficient demonstration on hardware

**Strengths And Weaknesses:**

Strength:

- The paper addresses an important problem of data efficiency and reset-free learning for real robotic applications.
- The paper clearly mentions the prior works on which the proposed system is based.
- Extensive experimentation to assess the effectiveness of the system both in simulated and physical environments.

Weakness:

The main weakness of the paper is that it is a bit difficult to see what the actual contribution of the paper is and how different the approach is compared to the previously published work. Several prior works already demonstrated how online RL can be accelerated with prior offline data.

**Summary Of Recommendation:**

in my opinion, the work is not significantly different from the prior work. However, the paper focuses on an important problem and demonstrated the effectiveness of their proposed approach through experimentation.

---

> ### Author Response · Authors · 2022-08-18
> **Author Response to Reviewer rVQq**
>
> Thank you for your thoughtful comments. Please see our top level response where we address your concerns of novelty. To provide further clarification, our contribution is not just to show that prior data makes RL more sample efficient, but to demonstrate how to build a complete system for incorporating prior data into robotic RL that simultaneously addresses sample-efficiency, zero-shot generalization, and autonomous non-episodic learning. This “novel combination of existing known modules,” in reviewer BwFL’s words, allows us to learn new tasks with infrequent resets where prior approaches for autonomous learning (R3L) fail completely. We hope that our system can act as a blueprint for future work that leverages prior data to make robotic RL more practical. We will revise to make this contribution more explicit. We will also update our references with the conference or journal versions.

---

### Official Review · Reviewer_CCLn · 2022-08-01

**Originality:** Fair
**Technical Quality:** Good
**Clarity Of Presentation:** Good
**Impact:** 3

**Recommendation:**

Weak Reject: I recommend rejecting the paper, but will not argue for my recommendation if the majority of other reviewers have a different opinion.

**Summary:**

The paper proposes to use a multi-task RL policy, trained offline with (i) data (state-action-state-reward tuples) for a collection of tasks, as a starting point to fine-tune, online, 2 new policies for a new, similar task (one policy to solve the task itself and one policy to autonomously reset the environment).
In case the new task is not sufficiently similar to the tasks for which the multi-task policy was pre-trained, (ii) additional data from operator-guided demonstrations are also used.
The approach is demonstrated both in real-world experiments and simulation for 6 pick-and-place tasks with a robotic manipulator.
The results in real-world experiments show how online fine-tuning improves the success rate of the multi-task policy on new tasks.
The paper interestingly brings together 3 ideas (offline learning, multi-task transfer learning, and learned automated reset policies) into a single approach.

**Issues:**

- I would recommend being very clear from early in the paper (e.g. Fig 1) and what is the motivation for the use of human demos and how they play into the learning pipeline, for what tasks, etc. (the explanation comes into multiple parts in sections 4 and 5)
- If extra space is needed (for clarification, additional results), Section 3 is very textbook-like and could be removed
- I would add a clarifying statement about frequent vs unfrequent and automated vs manual resets
- Provide some numerical figure when saying that the scripted policy fails "very frequently"
- Tables 1 and 2 should be improved, e.g. clarifying in the captions what the ratios stand for, separating in and out of distributions tasks, etc.
- line 300, should it be 360 instead of 600? what was the "offline only" zero-shot generalization performance?

**Quality Of The Limitations Section:**

Additional details required

**Reviewer Expertise:**

4: The reviewer is confident but not absolutely certain that the evaluation is correct

**Robotics Focus:**

Sufficient demonstration on hardware

**Strengths And Weaknesses:**

The paper makes simultaneous use of three very interesting and promising methods (offline learning, multi-task transfer learning, and learned automated reset policies), and treats the one-hot task representation of the multi-agent policy as a continuous, learnable parameter that also represents the task embedding of new tasks (and the reset policy), although not entirely new, is a neat approach to address the paper's problem.
Besides the marginal/incremental novelty, one possible weakness of the paper is that it seems that the new tasks need to be such (similar enough) that the old multi-task policy can already achieve success. The role of human demonstrations (initially presented as optional, then used for so-called out-of-distribution new tasks) should be better introduced and more formally justified.
The way data and results are presented in the tables should be improved for clarity and stand-alone readability

**Summary Of Recommendation:**

The paper is easy to read, enjoyable, and summarizes very interesting ideas for real-world robotics RL. My main concerns are the lack of novelty and the not overly clear presentation of the real-world experimental results.

---

> ### Author Response · Authors · 2022-08-18
> **Author Response to Reviewer CCLn**
>
> Thank you for your thoughtful review. We will first address your primary concerns about the novelty of the method and the clarity of presentation for the real-world results.
>
> While our method does combine existing ideas for offline RL, multi-task RL, and automated resets, we do not believe our contribution is incremental. Prior work has not explored initializing reset skills with prior data, and we find this enables novel capabilities. For instance, our method learns in limited reset settings where prior work (e.g., R3L) fails to make progress (see Figure 3). We believe impactful robotics research does not consist solely of proposing novel algorithms, but also involves integrating past work into complete systems and performing real-world experimentation. We also address novelty in our top level response. Please let us know if this addresses your concerns or if we can provide additional clarification.
>
> We agree that the clarity of the tables can be improved for stand-alone readability and have revised them in the PDF attached to the top level comment. We expanded the captions to fully explain the experiments that generated each set of results. In Table 1, we separated the tasks that use the direct transfer and transfer with demonstrations setting. In Table 2, we added a column to indicate which target task policy is used for each unseen task. Please let us know if the presentation is still confusing or if we can make additional revisions.
>
> Now, we will address several additional concerns brought up.
>
> We would like to correct a potential misunderstanding around whether we require the new tasks to be similar enough to the prior data that the pretrained policy achieves some success. This is not a requirement for our method *if demonstrations are provided*. In fact, we show that given a small number of demonstrations our method can learn several out-of-distribution tasks that are more significantly different from the prior data. For instance, our method can learn the task of inserting a ring onto a peg even though the prior data consists of only placing objects into containers. It is true that in our direct transfer setting, where no demonstrations are provided, we require that the policy trained on the prior data can achieve some success. This is because our tasks have sparse rewards, so if the policy never succeeds, it will never see a positive reward which makes fine-tuning in a reasonable amount of time infeasible. However, as we discuss in our top level response, the assumption that training and test tasks share structural similarity is necessary in all work on pretraining or transfer learning. Please let us know if we can provide any additional clarification on this topic.
>
> In regard to the role of demonstrations, we refer to them as “optional” not to mean that our method always works without demonstrations, but rather to indicate that the user can decide whether or not to include them. As explained above, we use demonstrations for tasks we call *out-of-distribution*, or tasks where a policy trained on the prior data alone will not achieve any success. In practice, the option to include demonstrations enables our method to adapt to tasks that are more significantly different from the prior data with only a small amount of additional effort. Importantly, we show that training on just the demonstrations is not enough to solve the task, meaning the agent must still transfer skills from the prior data. In the revised PDF, we have removed the term “optional” to avoid confusion and added additional explanation of the transfer with demonstrations setting in Section 4.
>
> In regard to our terminology around resets, a “manual/external” reset is when a reset is provided by human intervention whereas an “automated” reset is when the robot performs the reset itself. By “frequent resets'', we mean an external reset is provided after every trial whereas “infrequent resets” means an external reset is provided only after many trials. In our experiments we provide an external reset every 20-30 minutes in the real world and every hour in simulation. We will clarify this terminology earlier in the paper.
>
> We list in the Appendix that the success rate of the scripted policy is 0.35 for the Tray Container datasets, 0.93 for the Tray Drawer datasets, and 0.47 for the Kitchen dataset, but will move this information to the main text.
>
> Yes, line 300 in the original paper has a typo and should actually say “see 100 trials vs 600 trials in Table 1.” Here we’re making the point that fine-tuning improves the performance on the task being fine-tuned, but can reduce performance on other tasks (see 360 vs 600 trials in Table 2).
>
> As for the offline only performance in our zero-shot generalization experiments, our intention in these experiments was to test the generalization performance of policies after some amount of fine-tuning. The offline only column in Table 1 shows the performance of the pre-trained policy without fine-tuning.

---

> ### Author Response · Authors · 2022-08-23
> **Follow up on response**
>
> Please let us know if our response has addressed your concerns, or if we can provide any further clarification. We would be happy to discuss in more detail. Thank you!

---

> > ### Comment · Reviewer_CCLn · 2022-08-26
> > **Answer to authors' comment**
> >
> > Thank you for taking the time to review and address my comments. Some of my concerns regarding novelty vs. incremental advancement, tasks' similarity, and the role of demonstrations remains (but I understand could be perceived as subjective).
> > I appreciate the effort done in re-creating tables 1 and 2, the addition of a numerical success rate, and the correction of the typo: I think those changes do help to improve the presentation of the article.

---

### Author Response · Authors · 2022-08-18
**Author Response to Common Concerns: novelty and the similarity between training and test tasks**

We thank the reviewers for their thoughtful feedback. We’ve attached a revised version of our paper where we’ve addressed several comments (changes are highlighted in blue).

We’d first like to address two common criticisms. First, multiple reviewers brought up the concern that the method is not particularly novel because it combines existing ideas for offline RL, multi-task RL, and reset-free learning. However, combining existing components to enable novel capabilities characterizes a large portion of impactful robotics research, where progress requires not only novel algorithms, but also integrative efforts, system building, and real-world experimentation. To our knowledge, prior work has not explored initializing reset skills for a new task with prior data for other tasks as we do. Our contribution is not necessarily to present a completely novel method, but to empirically demonstrate that properly incorporating prior data makes robotic RL easier in several ways: improving sample-efficiency, providing reset skills, and enabling broad generalization.

Multiple reviewers were also concerned that the test tasks need to be similar enough to the tasks in the prior data that a policy trained on the prior data achieves a non-zero success rate. First, we note that we only make this assumption in the more difficult direct transfer setting where the robot must solve an unseen task with sparse reward and without any demonstrations. In this setting, if the robot never succeeds at the task, it will never achieve states with positive reward, which makes fine-tuning in a reasonable amount of time infeasible in the real world. We also study the transfer with demonstrations setting where the user provides a small number of demonstrations of the test task. We show that in this setting our method can fine-tune to tasks that are different enough from the prior data where training on the prior data alone would not result in any success. For instance, we fine-tune to the task of inserting a ring onto a peg even though the prior data consists of only placing objects into containers. Thus, our method is not limited to solving tasks that are highly similar to those in the prior data, though of course some structural similarity needs to exist between the prior tasks and new task in order for the offline data to provide any benefit over training from scratch. As discussed in our limitations section, we acknowledge that without access to demonstrations, the new tasks need to share significant structural similarity with the prior data. Indeed, some assumption like this is needed for any transfer learning or pretraining method. But we believe that as the breadth and diversity of the prior data increases, the pretrained policy will be useful for a broader range of tasks.

Please let us know if this addresses your concerns. We will also respond to more specific reviewer points individually.

---

### Meta-Review · Area_Chair_u8et · 2022-08-04

**Recommendation:** Accept (Poster)
**Confidence:** 3

**Metareview:**

Please check the comments of the reviewers in detail.

### Strengths
- interesting combination of existing ideas
- experiments with a real system (low-cost arm)

### Weaknesses
- not highly novel: combination of existing ideas
- comparison with the state-of-the-art could be stronger: using prior data has been explored several times in the past
- statistics need to be clarified (number of replicates, including for the real-world experiments, evaluation of variance, etc.)
- assumption that both forward and backward policies can be learned with a few random shoots, which seems unlikely in most scenarios

#### Post-rebuttal remarks
I would like to thank the authors for their efforts in answering the comments. The main concern left is the novelty & potential impact, but, overall, the reviewers liked the paper.

---

> ### Author Response · Authors · 2022-08-18
> **Author Response to Area Chair u8et**
>
> Thank you for your summary. Please see our top level comment above where we address concerns of novelty.
>
> We’d also like to clarify that we do not assume that the forward and backward policies for a new task can be learned with a few trials (we use up to 600 trials in our real world experiments). We do assume, in a setting without demonstrations, that our pretrained policy can achieve a non-zero success rate over the course of many trials (greater than 30). We address this concern in the top level response above. To summarize, we make this assumption because our tasks have sparse rewards and if the agent does not succeed at all over many trials, fine-tuning will be prohibitively slow in the real world. However, the applicability of our method is not entirely limited by this assumption because our method can also use a small number of demonstrations, which we show enables learning tasks that are more significantly different from the prior data.
>
> We have added further details of the statistics of our experiments to the paper. The results for our simulated experiments are averaged over 4 random seeds and our plots show confidence intervals. The success rates for our real world experiments are calculated over 10 trials.